# Desktop Micro-EDM System for High-Aspect Ratio Micro-Hole Drilling in Tungsten Cemented Carbide by Cut-Side Micro-Tool

**DOI:** 10.3390/mi11070675

**Published:** 2020-07-11

**Authors:** Yung-Yi Wu, Tzu-Wei Huang, Dong-Yea Sheu

**Affiliations:** 1Graduate Institute of Mechanical & Electrical Engineering, CMEE, National Taipei University of Technology, Taipei 10608, Taiwan; wusiryy@gmail.com; 2Graduate Institute of Manufacturing Technology, CMEE, National Taipei University of Technology, Taipei 10608, Taiwan; aspz519930@gmail.com

**Keywords:** Tungsten cemented carbide (WC-Co), desktop micro-electrical discharge machining (micro-EDM) system, cut-side micro-tool, micro-holes

## Abstract

Tungsten cemented carbide (WC-Co) is a widely applied material in micro-hole drilling, such as in suction nozzles, injection nozzles, and wire drawing dies, owing to its high wear resistance and hardness. Since the development of wire-electro-discharge grinding (WEDG) technology, the micro-electrical discharge machining (micro-EDM) has been excellent in the process of fabricating micro-holes in WC-Co material. Even though high-quality micro-holes can be drilled by micro-EDM, it is still limited in large-scale production, due to the electrode tool wear caused during the process. In addition, the high cost of precision micro-EDM is also a limitation for WC-Co micro-hole drilling. This study aimed to develop a low-cost desktop micro-EDM system for fabricating micro-holes in tungsten cemented carbide materials. Taking advantage of commercial micro tools in a desktop micro-EDM system, it is possible to reach half the amount of large-scale production of micro-holes. Meanwhile, it is difficult to drill the deep and high aspect ratio micro-holes using conventional micro-EDM, therefore, a cut-side micro-tool shaped for micro-EDM system drilling was exploited in this study. The results show that micro-holes with a diameter of 0.07 mm and thickness of 1.0 mm could be drilled completely by cut-side micro-tools. The roundness of the holes were approximately 0.001 mm and the aspect ratio was close to 15.

## 1. Introduction

Tungsten carbide (WC) is a materiel which is widely applied in military industrial composite, metallurgy, aerospace, and other important fields because of its excellent physical and chemical properties [1]. Pure WC is very brittle. If doped with small amounts of titanium, cobalt, or other metals, the incorporated brittleness can be reduced. The interaction between Co-based binders and WC grains on early stages of liquid-phase sintering can be strongly affected by the carbon content in the binders [2]. Tungsten cemented carbide (WC-Co) has a series of excellent properties, such as hardness, strength, toughness, wear resistance, and corrosion resistance [3,4]. 

Micro-holes made from WC-Co are widely used as spraying nozzles, injection nozzles, and spinning nozzles, owing to their low wear and hardness [5]. Machining processes, such as micro-mechanical drilling, laser machining (LBM), and electron beam machining (EBM), are typically used for the mass- or semi-mass production of micro-holes in WC-Co materials. The micro-electrical discharge machining (micro-EDM) process is highly suitable for micro-hole fabrication because it is burr-free and efficient irrespective of the workpiece hardness, especially since the development of the WEDG technology [6,7,8]. However, micro-EDM still has some limitations for the mass production of micro-holes due to the low productivity of the micro-tools used for fabrication [9,10].

In order to achieve semi-mass production of micro-hole drilling using EDM in WC-Co material, a low-cost desktop micro-EDM system was developed in this study. Off-the-shelf spindle electrode tools with diameters of 0.15 mm were used directly as microelectrode tools. These tools have proven to be commercially successful in low-cost mechanical drilling. Using these commercially available micro-spindle tools, it was possible to achieve the semi-mass production of a WC-Co material with micro-holes drilled via a desktop micro-EDM system. However, small diameter and long electrode tools with diameters less than 0.1 mm are not commercially available. Therefore, in this study, a WEDG unit was attached to the desktop micro-EDM system to fabricate micro-spindle tools with diameters less than 0.01 mm by using commercially available 0.15 mm tools. In order to produce micro-holes with a high aspect ratio, a single-side notch electrode method was applied to flush debris. The machining parameters, such as machining time, aspect ratio, spindle tool wear, and micro-hole quality, were investigated in this paper. It is expected that the desktop micro-EDM system will be potentially useful for drilling micro-holes in tungsten carbide materials.

## 2. Structure of the Desktop Micro-EDM

### 2.1. Desktop Micro-EDM Structure

The micro-EDM with three axis computer numerical control (CNC) controllers has been commercialized in the industry market [11]. The high accuracy controlling system makes micro-EDM more expensive. However, most micro-EDM systems are still designed for micro-holes drilling in industry applications. Low productivity of micro-holes drilling is still the main challenge for micro-EDM due to the significant tool wear and micro-tools fabrication [12,13]. To achieve mass-production of micro-hole fabrication, these two factors need to be addressed in this paper.

The desktop micro-EDM system was developed and designed in this research. The block diagram of the operation relationship of each unit is shown in Figure 1. The system has only three axes, X, Y, Z for the micro-hole drilling, as shown in Figure 2. The X-Y stages were controlled manually with digital indicators, and the Z-axis was controlled by the microcontroller unit (MCU). The most important components of the desktop EDM system were the V-shaped block and the spindle electrode tools. The structure of the spindle tool is shown in Figure 3. The spindle tool was mounted on the V-shaped block and rotated by a direct current (DC) motor. The rotation speed was variable. The linear straightness and roundness of the spindles were important to ensure highly accurate micro-hole drilling. This machine was mainly suitable for micro-holes with diameters of 0.15 mm or less with micro resistor capacity electro discharge circuit (RC circuit). In this paper, the desktop EDM was designed for micro-holes drilling with diameters less than 0.15 mm. The desktop micro-EDM system was available for electro-conductive materials and used conductive materials such as tungsten (W), tungsten cemented carbide (WC), die steel (SKD), and stainless steel (SUS). The discharge energy of the desktop micro-EDM system simply adopts a RC discharge circuit with DC power supply of 80 to 100 V, and the workpiece cannot be touched during discharge machining. The discharge power must be turned off when replacing the microelectrode tool. The main specifications of the desktop micro-EDM system is shown in Table 1 [14].

### 2.2. Commercially Available Spindle Tools

The desktop micro-EDM system uses off-the-shelf spindle electrode tools. The micro-tool length and diameter are 5 mm and 0.15 mm, respectively, as shown in Figure 4. It was successful and commercially available for the mass production of electrode tools and had a shank diameter of 3.0 mm and no screw slots due to the mechanical grinding process. The micro-tool accuracy of both the diameter and the length was approximately 0.003 mm for tungsten cemented carbide. Figure 5 shows the spindle micro-tool with pulley [15]. The pulley was fixed onto the spindle tool in the system, and it rotated directly on the V-shaped block. The rotation roundness of spindle tool was approximately 0.5 μm without any vibration. Due to a low tool wear ratio of the tungsten carbide by micro-EDM, the tool electrode is not available for commercial polishing process if it is made by tungsten. The WC-Co material demonstrates the possibility of mass production of the micro-tool by mechanical grinding process [16]. Even though it was possible to fabricate spindle micro-tools with a diameter of 0.05 mm by conventional grinding process, the aspect ratio was only 3 or 4 due to the mechanical grinding force. This low aspect ratio of spindle tools makes them unsuitable for mass-drilling micro-holes by micro-EDM. The other critical challenge involved in grinding is the fabrication of micro-tools with diameters less than 0.03 mm. In this paper, WEDG technology was used for micro-tools fabrication with diameters less than 0.03 mm by using commercial tools directly for high aspect ratio micro-hole drilling.

### 2.3. WEDG Technology Unit

As described in the previous section, mechanical grinding process has been successful for the mass production of micro-spindle tools. The diameter was approximately 0.003 mm. However, it is still difficult to fabricate spindle tools with diameters less than 0.1 mm by using the grinding process, due to the mechanical grinding force [17]. It is possible to produce small spindle tools with a high precision grinding machine, however, the aspect ratio is only 3 or 4 with diameters less than 0.1 mm. Hence, in this study, in order to fabricate ultra-micro-holes with diameters of less than 0.05 mm, the wire-electro-discharge grinding (WEDG) technology unit was attached to the desktop micro-EDM system, as shown in Figure 6 [18,19]. The off-the-shelf spindle tools with diameters of 0.15 mm and lengths of 5 mm from the commercial market were used directly. The material of the spindle tool was tungsten carbide, which provides sufficient toughness and rigidness. Compared to the conventional WEDG process, the micro-electrode tool fabrication technique employed in this study will be more efficient, due to the finishing process post-machining. The x-axis was manually controlled by a digital micrometer head with the position control display resolutions as low as 1 μm, as shown in Figure 7. By aligning the position of the x-axis, the micro-electrode tools could easily be fabricated using WEDG technology. It is thus possible to shape the tools through one machining process only. 

### 2.4. Control Pad Micro-Controller Chips (dsPIC)

The conventional micro-EDM system is usually operated by numerical control (NC) or computer numerical control (CNC) controllers [20]. In addition, the position alignment and the scanning process with tool compensation are also possible. However, the large cost of conventional micro-EDM systems makes EDM unpopular for micro-hole drilling. To reduce the cost, this study used the MCU digital signal peripheral interface controller (dsPIC), to control the movement of the spindle tool and to detect the discharge gap [21]. The I/O connection provides drilling depth selection. The z-axis of the spindle micro-tool feeding rate is controlled by detecting voltage of the gap discharge. By only pushing the start button, the desktop micro-EDM system can automatically produce micro-holes. The diameter of the microelectrode tool could be manually adjusted using the x-axis position alignment with high resolution indicator. The micro-tools and micro-hole fabrication can be carried out on the same desktop micro-EDM system. Complete internal structure of the desktop micro-EDM system is as shown in Figure 8.

## 3. Micro-Holes Drilling by Desktop Micro-EDM System

### 3.1. Micro-Electrode Tool Fabrication by WEDG

In this study, two types of micro-spindle tools with diameters 0.15 mm and 0.035 mm were used to fabricate the WC-Co micro-holes by micro-EDM drilling. To achieve the semi-mass production of micro-hole drilling using micro-EDM, commercial micro-spindle tools with a diameter of 0.15 mm can be used directly, but there is no supply for diameters less than 0.050 mm. However, fine micro-spindle tools may be produced by the WEDG grinding process, as shown in Figure 5. In order to fabricate micro-tools with diameters less than 0.035 mm, the WEDG unit mounted on the table was still used to reform the commercial spindle tools using only the finishing process. Figure 8 shows the potential of using the desktop EDM system to fabricate micro-tools with diameters of only 0.007 mm with tungsten carbide, as shown in Figure 9. It is thus possible to fabricate micro-holes with diameters less than 0.01 mm using this low-cost desktop micro-EDM system. By using commercial tools directly, the desktop EDM is able to produce micro-spindle tools more efficiently without rough machining.

### 3.2. Micro-Hole Drilling by Micro-EDM System

The desktop micro-EDM system uses an RC discharge circuit and DC power with an open voltage of 80 V [22]. The machining conditions of the desktop micro-EDM system is as shown in Table 2. The V-shaped block was mounted onto the desktop micro-EDM system with high accuracy. Commercial micro-tools were used directly without the WEDG technology and the electrode tools are available from commercial market. However, for micro-tools with diameters less than 50 μm, the WEDG technology was still applied for reforming in this study. The main characteristic of the low-cost desktop micro-EDM system is that micro-tool and micro-hole convenience fabrication can be achieved conveniently on the same machine. After shaping the microelectrode tools, it is possible to drill micro-holes directly on the same micro-EDM. Experimental results of micro-hole drilling using tools with diameters of 150 μm and 35 μm are shown in Figure 10. It is facile to drill micro-holes on the WC-Co using this desktop micro-EDM system without any burr. It takes less than 2 min to drill a micro-hole of 150 μm diameter by the 1000 pf capacitor; and it takes approximately 5 min to drill a micro-hole of 40 μm when using discharge capacitor of 100 pf. Thus, when conducting less discharge capacitance, the machining time takes longer, and the electrode wear increases. Therefore, a small capacitance value will result in optimal micro-holes machining. 

The surface observation of the micro-holes by Scanning Electron Microscope (SEM) is shown in Figure 11 [23]. Compared to another machining process, the desktop micro-EDM process can meet the requirement of micro-holes drilling on WC-Co material without any burrs. The experimental results show micro-tools with diameters less than 0.01 mm and micro-holes with diameters less than 0.05 mm. In micro-hole machining, the horizontal diameter (Dx) for a 0.3 mm thick workpiece is 39 μm, and the vertical diameter (Dy) is 40 μm; whereas the horizontal diameter (Dx) for 0.5 mm, the thick workpiece, is 66 μm, and the vertical diameter Dy is 66 μm. Therefore, the machining roundness is approximately 1 μm. For machining workpieces with apertures of 40 and 66 μm and thicknesses of 0.3 and 0.5 mm, the aspect ratio is about 8.

### 3.3. Electrode Tool Wear and Roundness

Figure 12 shows the length of tool wear of EDM micro-holes drilling with different electrical capacites. The micro-tools maintain their shape with only little tool wear by using small electric discharge capacities [24]. However, the tool wear increases significantly when the diameter is smaller than 0.05 mm. Even though the larger electric discharge capacities could increase the efficiency of micro-hole drilling, it leads to greater tool wear and deterioration of the micro-hole roundness. Therefore, it is necessary to consider all parameters to identify a suitable discharge capacity. In this study, for a work piece thickness of 0.3 mm and tool electrode diameter of 150 μm, approximately 1000 pF is the ideal drilling capacitance, whereas, for a tool electrode diameter of approximately 35 μm, a capacitance of 100 pF is the optimal value for micro-hole drilling.

### 3.4. Limitations of the Machining Depth

Normally, transistor discharge circuits are used in commercial EDM because the discharge energy is controllable by adjusting the pulse generator and duty cycle [25]. However, micro-EDM requires extremely small electric discharge energy, especially for micro-tools with diameters less than 50 μm. The resistor capacity (RC) pulse generator is popular and widely used for the micro-EDM electric discharge energy [26]. The discharge energy depends on the capacity of the RC pulse generator. Therefore, the main parameter of the RC discharge circuit is the magnitude of capacity. Theoretically, it is possible to increase the machining speed with large electric capacity due to the large discharge energy. The large capacitors could cause bigger discharge sparks, however, the large discharge energy will lead to significant tool wear. The high aspect ratio of micro-holes will not be drilled to penetrate completely as the micro-tool wear will exceed the feed depth. As shown in Figure 13, the spindle feeding speed is efficient without any stagnation while the feeding depth is below 800 μm. However, the machining speed decreases significantly when the spindle feeding depth is larger than 800 μm due to insufficient debris flushing. Even the spindle feeding depth increases. However, this phenomenon means a large amount of micro-tool wear. It is clear that the micro-EDM drilling process is a method to fabricate micro-holes with aspect ratios less than 5 exclusively. The high aspect ratio micro-spindle tools are not possible for deep micro-hole drilling using micro-EDM due to the significant tool wear [27].

### 3.5. High-Aspect Micro-Holes Fabrication

Micro-EDM encounters another critical problem for high-aspect ratio on deep micro-holes machining, especially when the diameter is less than 100 μm and the aspect ratio is larger than 10 [28]. The cut-side shaped micro-electrode tool was capable of fabrication using the desktop micro-EDM system. The purpose is to enhance the machining efficiency for micro-holes drilling with high aspect ratio. The 50 μm diameter tool-electrode can be ground out to 10 μm in depth by the WEDG technology machining unit without any tool spindle rotation, shown as in Figure 14. The side view and front view of cut-side shaped micro-tool after the WEDG process is shown as in Figure 15 and Figure 16. The cut-side electrode tools of special shapes are able to improve the debris removal problem, but there is still no elevation for higher aspect ratio due to high tool wear [29,30]. Figure 17 shows the comparison between the feeding depth of the cylinder tool and cut-side tool. The experiment shows that initially the smaller cut-side electrode brought more electrode wear than the bigger cylindrical electrode. Later, the cut-side electrode contrarily brought less electrode wear than the bigger cylindrical electrode, for its larger space removed debris faster. Under such circumstances, to reduce high tools wear and increase machining efficiency of micro-holes EDM drilling, the cylinder and cut-side shaped micro-electrode tool should be shifted alternately for deep micro-holes drilling. In micro-holes machining with high-aspect ratio, the feeding depth was drilled alternately by cylindrical and cut-side micro-electrode tools. At first, the cylindrical micro-tool with less electrode wear started to drill micro-holes to reach the feeding depth limit area, and then shifted to the cut-side electrode with larger space to process it at the same speed without moving the workpiece. Continuing the process until the micro-holes completely drilled might improve the machining efficiency. The best machining method was achieved by using the cylinder- and cut-side shaped micro-tools alternately; a micro-hole with high aspect ratio will be drilled completely at 15 times larger in about 10 min of the machining time. The results show that it can be completely drilled by cut-side micro-tools, and high aspect ratio drilling can be performed on WC-Co material with a thickness of 1.0 mm, with an inlet diameter of 73 μm and an outlet diameter of 58 μm, as shown in Figure 18. The micro-holes surface with high aspect ratio on WC-Co was observed through SEM, as shown in Figure 19.

## 4. Conclusions

A review of the research trends in micro-EDM about various electrode tools and their effects in the characteristics of micro-EDM are presented. A low-cost desktop micro-EDM system was explored and developed for rapid drilling of micro-holes through tungsten cemented carbide in this study. Using commercial electrode tools of 50 up to 150 μm, for about 2 to 4 min, it is possible to achieve semi-large-scale production of micro-holes. The desktop micro-EDM system is also able to drill micro-holes by mechanical drilling using a micro-spindle tool with a screw slot. Besides, the WEDG technology can even be employed for more meticulous micro-electrode and shaping tool fabrication. In addition to superb microelectrode tool fabrication, micro-hole drilling could be also done by the same machine. Compared to the nearly one million dollars and high prices of commercial micro-EDM systems, there are more potential applications for the drilling of tungsten cemented carbide in the low-cost desktop micro-EDM system developed in this study, which is able to produce roundness of a micro-hole of approximately 1 μm and a 9 times standard aspect ratio, enhanced to 15 times in high-aspect ratio. We hope that research and analyses on machining characteristics of double cut-side tool electrodes will be continued in the future.

## Figures and Tables

**Figure 1 micromachines-11-00675-f001:**
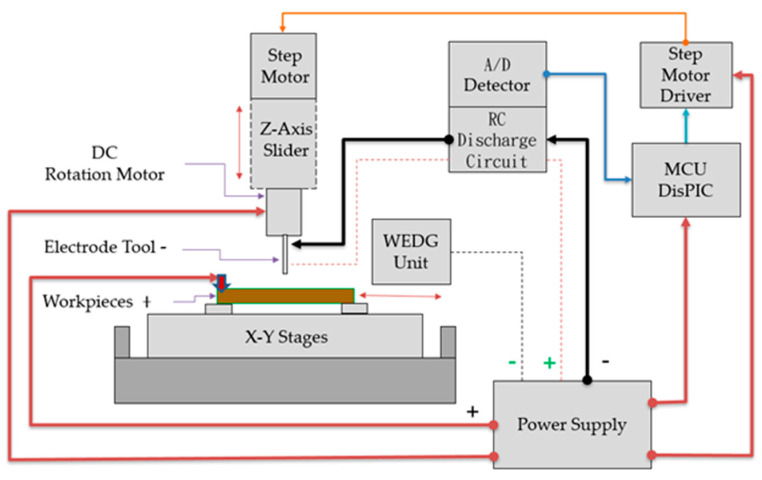
The block diagram sketch of the desktop micro-electrical discharge machining (micro-EDM) system.

**Figure 2 micromachines-11-00675-f002:**
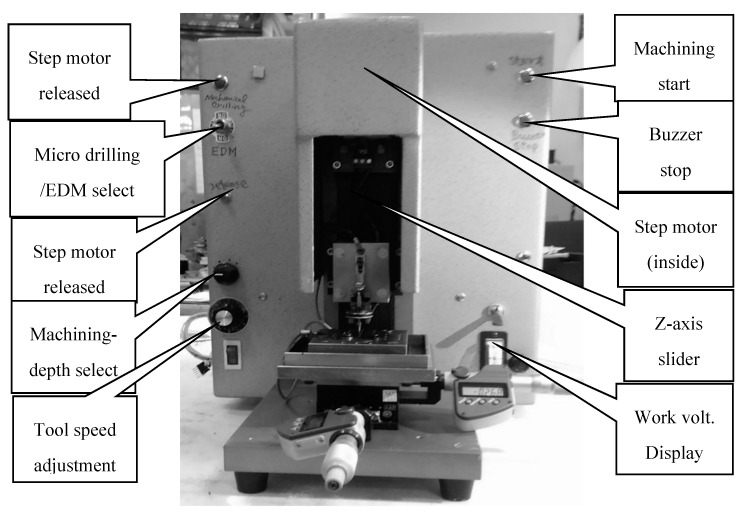
Complete structure of the desktop micro-EDM system.

**Figure 3 micromachines-11-00675-f003:**
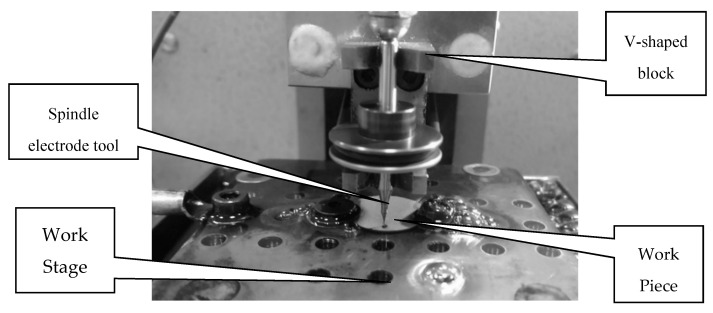
Structure of the spindle tools on the V-shaped block.

**Figure 4 micromachines-11-00675-f004:**
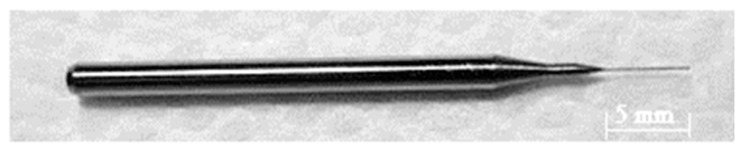
Commercially available spindle micro-tool without the screw slot. Diameter of shank: 3 mm. Diameter of tool: 0.15 mm.

**Figure 5 micromachines-11-00675-f005:**
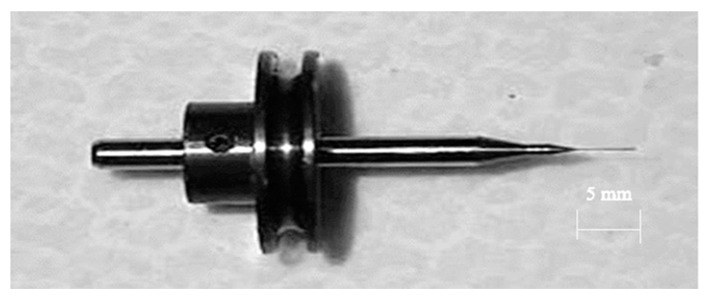
Spindle micro-tool with pulley.

**Figure 6 micromachines-11-00675-f006:**
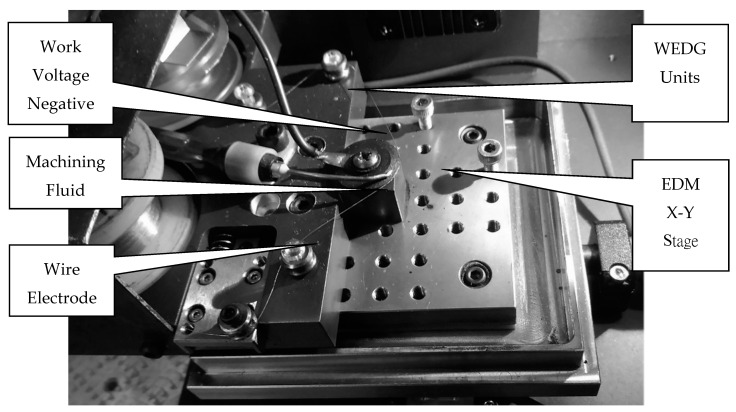
Wire-electro-discharge grinding (WEDG) technology unit attached to the desktop micro-EDM system.

**Figure 7 micromachines-11-00675-f007:**
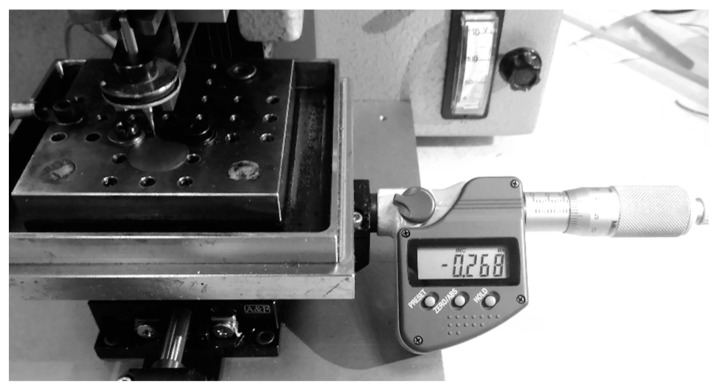
X direction manual control by digital micrometer head.

**Figure 8 micromachines-11-00675-f008:**
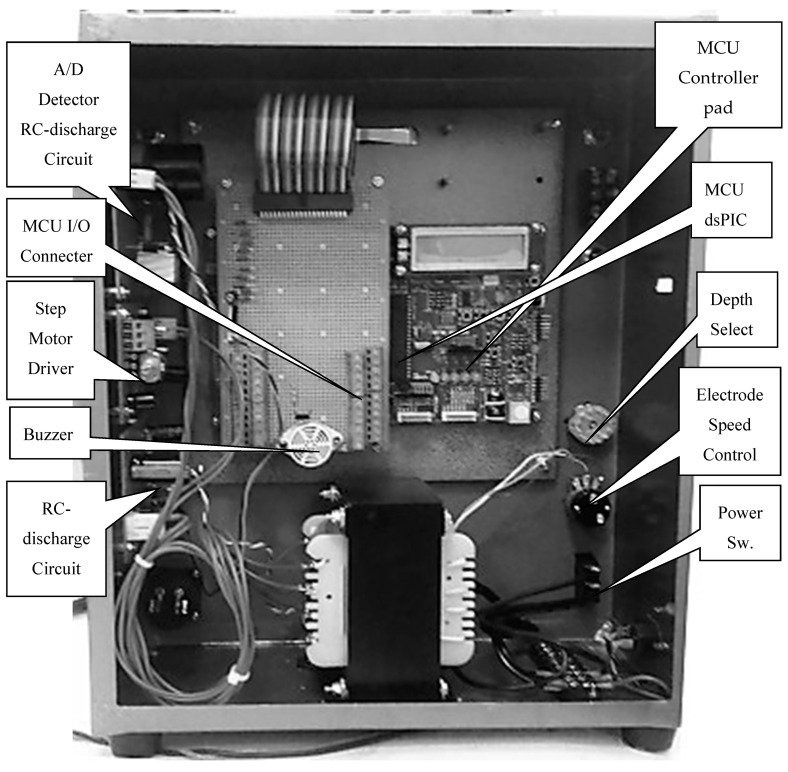
Internal structure of the desktop micro-EDM system.

**Figure 9 micromachines-11-00675-f009:**
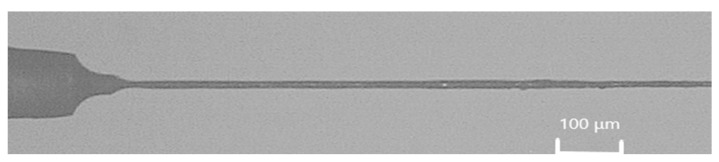
Microelectrode tool with diameter of 0.007 mm.

**Figure 10 micromachines-11-00675-f010:**
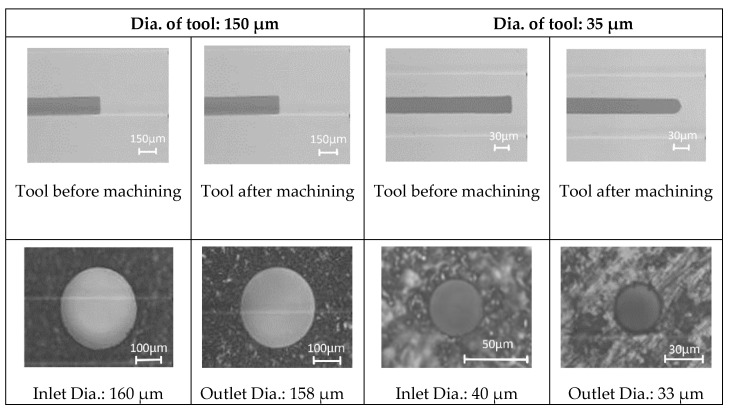
Micro-hole drilling on tungsten cemented carbide (WC-Co) through a desktop micro-EDM system.

**Figure 11 micromachines-11-00675-f011:**
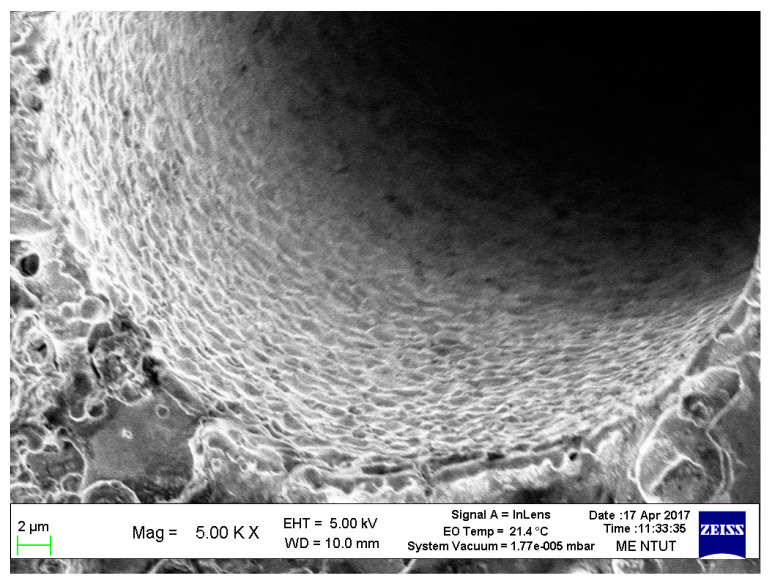
Surface of WC-Co micro-holes observed by Scanning Electron Microscope (SEM).

**Figure 12 micromachines-11-00675-f012:**
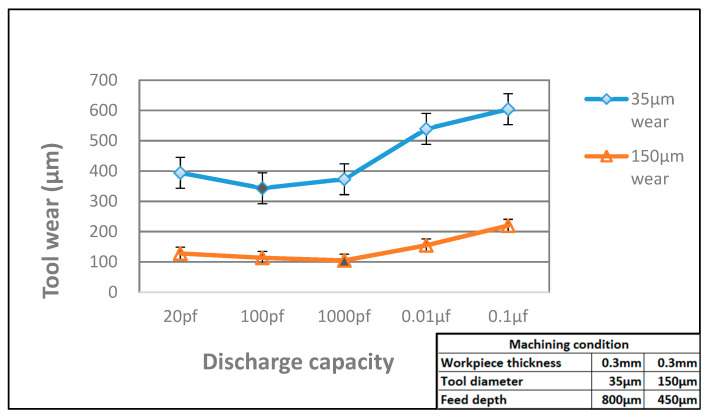
Tool wear in micro-hole drilling of WC-Co.

**Figure 13 micromachines-11-00675-f013:**
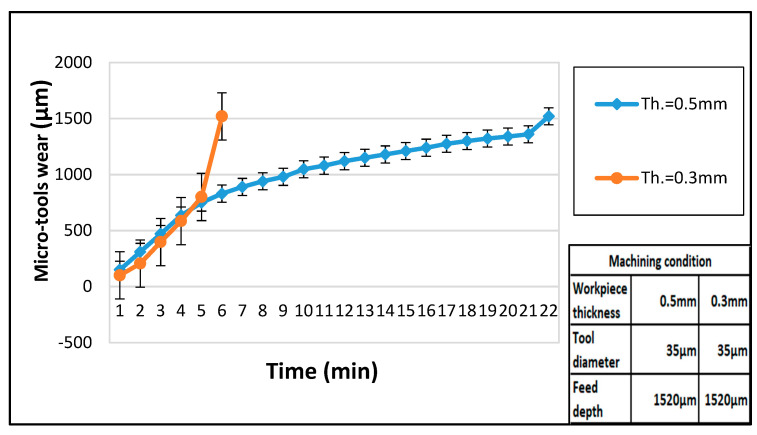
Micro-tools wear on WC-Co by the desktop micro-EDM.

**Figure 14 micromachines-11-00675-f014:**
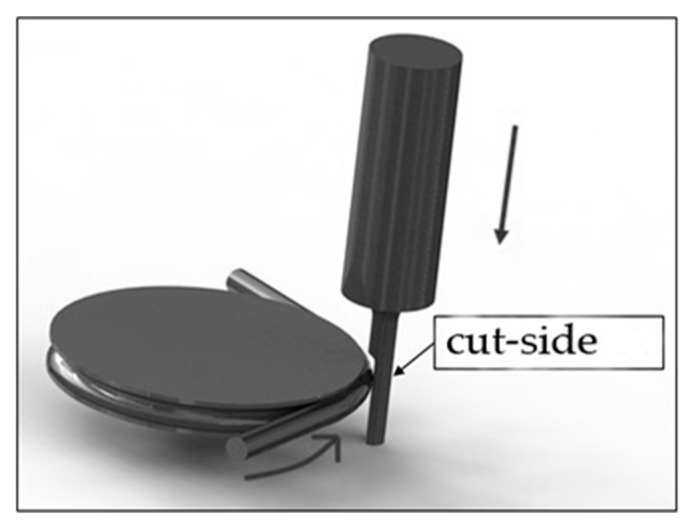
Cut-side shaped micro-tool process by WEDG.

**Figure 15 micromachines-11-00675-f015:**
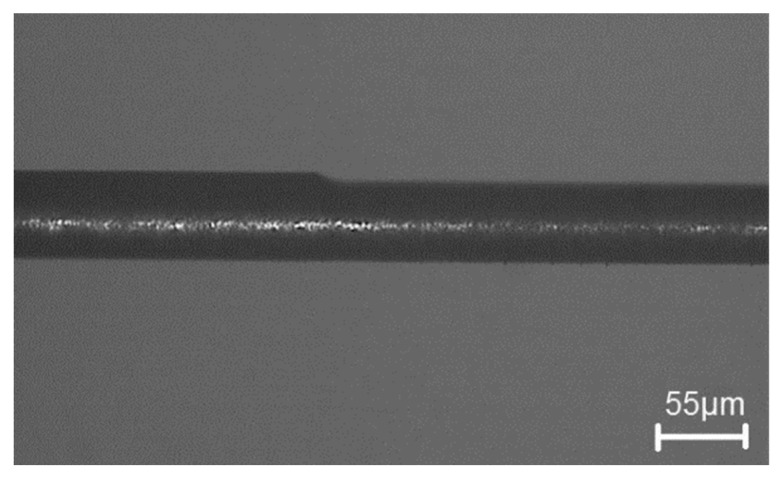
Side view of cut-side shaped micro-tool.

**Figure 16 micromachines-11-00675-f016:**
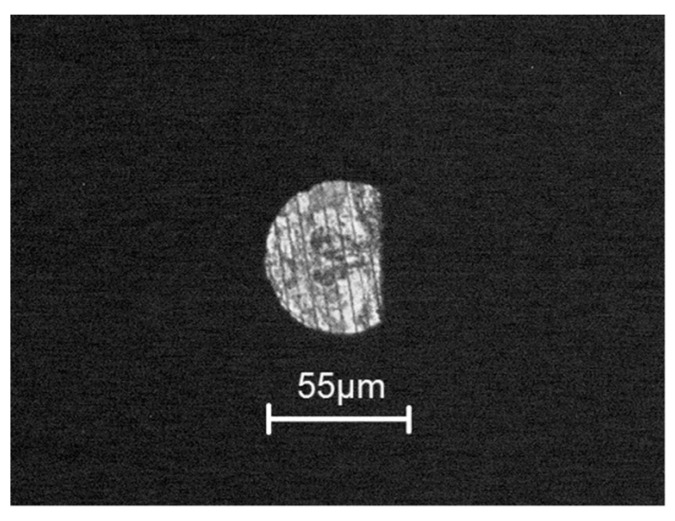
Front view of cut-side shaped micro-tool.

**Figure 17 micromachines-11-00675-f017:**
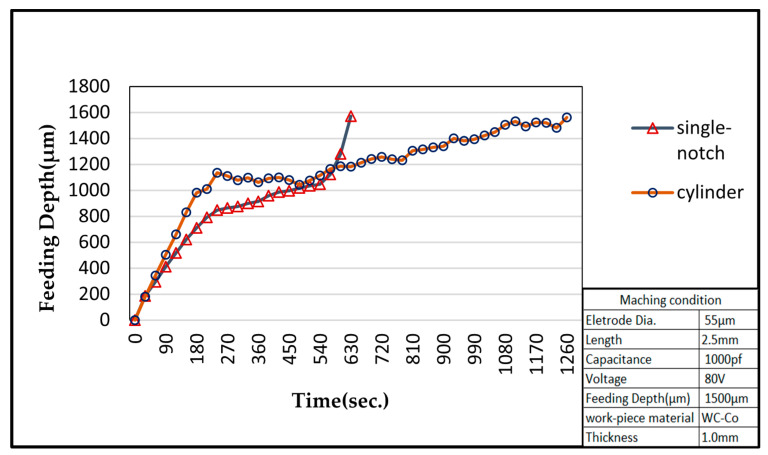
Comparison between the feeding depth of the cylinder tool and cut-side tool.

**Figure 18 micromachines-11-00675-f018:**
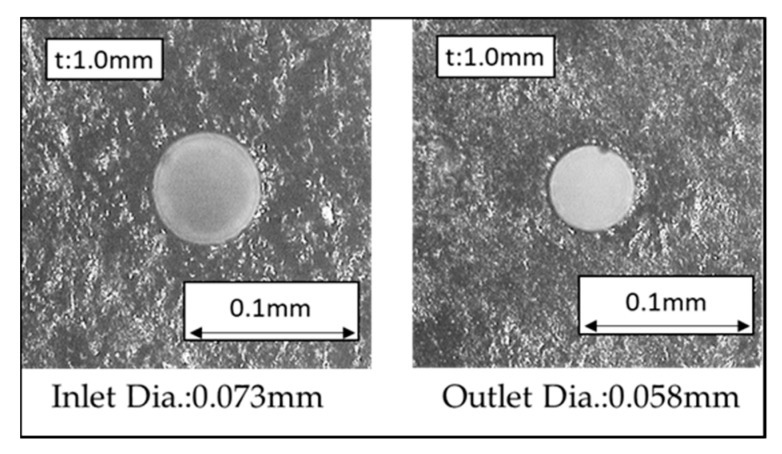
Micro-hole drilling of high aspect ratio on WC-Co through a desktop micro-EDM system.

**Figure 19 micromachines-11-00675-f019:**
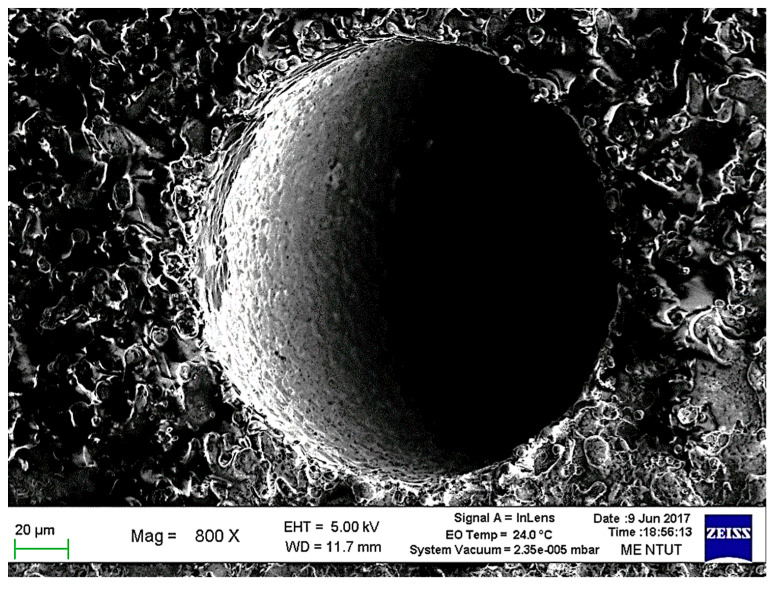
Surface of micro-hole drilling of high aspect ratio on WC-Co observed by SEM.

**Table 1 micromachines-11-00675-t001:** Main specification of the desktop micro-EDM system.

Desktop Micro-EDM System Specification		
Machining dimensions (L × W × H)	450 mm × 320 mm × 360 mm	
Total weight	5 kg	
XY stage (L × W)	300 mm × 200 mm	
X, Y travel	25 mm × 25 mm	by manual process
Z-axis travel	150 mm	by microcontroller unit (MCU) control
Z-axis resolution	0.1 μm	By step motor
Spindle speed	6000 RPM	
Electrode diameter	35 up to 150 μm	

**Table 2 micromachines-11-00675-t002:** Machining conditions of the desktop micro-EDM system.

Machining type	RC-EDM
Tool material	WC (150 μm)
Workpiece material	WC-Co
Machining open voltage	80 V
Discharge resistor	500 Ω × 2
Discharge capacitor	100 or 1000 pf
Workpiece thickness	0.5 or 0.3 mm
Machining depth	800 or 450 μm
Tool diameter by standard offer	150 μm
Tool diameter grind by WEDG	35 μm

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
