# Peer review of "Desktop Micro-EDM System for High-Aspect Ratio Micro-Hole Drilling in Tungsten Cemented Carbide by Cut-Side Micro-Tool"

_micromachines, 2020, doi:10.3390/mi11070675_

Round 1

Reviewer 1 Report

The authors present a desktop micro-EDM tool for machining of small and deep holes in cemented WC. Following comments need to be incorporated in the manuscript.

  1. The authors claim roundness of 0.001 mm. How did the authors measured roundness
  2. Fig. 1. The picture gives no information and looks out of proportion. Please add a sketch depicting the major peripherals of the machine.
  3. Add scale bar in Fig. 3, 4.
  4. Figure 5 and Figure 7 should be properly labelled or add a sketch. This will be more informative for the readers.
  5. In the abstract the authors claim 70 um diameter holes but in Fig. 9, it can be seen that either the diameters are more than 100 um or less than 50um. This is conflicting with what authors state in abstract.
  6. Figure 11, 12: Add the error bars to show the deviation of your measurements.

Author Response

Dear reviewers,

Thank you for your patience in reviewing and giving comments for corrections of my paper. In respond to your advices, my revision is submitting as follows:

  1. Regarding the question of how to measure roundness: It is calculated based on the difference between the horizontal diameter and the vertical diameter. This has been revised and added from line 189.
  2. The Figure 1 has been revised and been changed number 1 into number 2. The sketch also has been added in Figure 1 as your comment.
  3. Scale bars have been added in Figure 4 and Figure 5 (the original Figures 3 and 4).
  4. Figures 5 and 7 have been labelled and been changed number to 6 and 8, with regard to adding sketches, it shows in Figure 1.
  5. In Fig. 9(10), the diameters are more than 100 um or less than 50um, which is to illustrate the possibility on commercial electrodes and WEDG-grind electrodes being used and completed for basic drilling machining in this micro-ECM system. I am sorry for my negligence for this point where I did not present the single-cut edge electrode at a depth of 1mm with a depth-to-diameter ratio of 70μm for drilling. Thank you very much for reminding of me. Figures 18 and 19 have been complemented.
  6. The Error bars of Figures 11 and 12 have been added.

Thank you for your precious comments and feedbacks.

I shall be appreciated if you can accept my revision and look forward to hearing from you soon

Sincerely yours,

Yung-yi WU

Taiwan

Reviewer 2 Report

The paper is well written and it has more technical soundness rather than merely scientific one. In my viewpoint, this work could be improved by including scientific explanations to the experimental results.  I might recommend the article for acceptance, but only if the authors address properly the following comments. 

Comments that must be addressed: Concerning the development of the desktop micro-EDM, it is not clear at all where the authors put their actual effort: they modified the spindle and spindle positioning, but what about the pulse generator? What is the utility of the pulley when using Wire-EDM? What do the authors mean when they stated: “Compared to conventional WEDG process, micro-electrode tool fabrication technique employed in this study will be more efficient due to the finishing process post machining”? Moreover, it does not seem that this statement has been proven after results discussion. I suggest the authors to underline why they state that this desktop micro-EDM is costly advantageous: do they mean in terms of machining time compared to quality of the drilled micro-holes? If so, I suggest them to include a table of comparison among their results and those already reported in the state of the art. The authors must also add information about flushing: what kind of fluid did they use? The authors must define tool wear: I suppose it is tool wear length, but no definition is provided. The authors must provide a physical explanation on why the increase of spindle feeding depth does not induce any improvement in the micro-hole machining and induce “only an increase of tool wear”. I suggest the authors to add scientific comments, also to give value to their work. 

Other comments:

Page 2 line 84: check sentence “attention to the workpiece…”; line 86: check punctuation.

Page 4: line 101-102: Please check sentence “Even the tungsten…”.

Page 10: line 239-244: these sentence of the micro-EDM drilling procedure by using cut-side tools is by no means clear. I suggest the authors to rewrite more explicitly this part. It is not clear why the authors adopted this micro-tool shape, too.

Last section: please rename it with “conclusion” rather than results.

Most of the desktop micro-EDM machine figures have low quality.

Please check captions of Figure 12: it states “desktop micro-ecm”.

Author Response

Dear reviewers,

Thank you for your patience in reviewing and giving comments for corrections of my paper. In respond to your advices, my revision is submitting as follows:

For the part of Comments:

  1. In this study the pulse generator is formed by a simple RC discharge circuit in the micro-EDM system, when it goes to the threshold, the charge and discharge processing occurs, where was explained in line 167.
  2. Pulley of WEDG provides the refresh negative copper wire immediately and the new copper wire maintains the processing quality. The additional Micro-EDM WEDG UNIT can provide fast and effective services of micro electrodes, such as the rapid manufacture of micro electrodes of 35um, special-shaped cut-side electrodes, to process the micro-hole drilling and high aspect ratio machining.
  3. Besides of the advantages of micro-EDM of low cost, there are some advantages such as the convenience for machining, tool electrode replacement, ability to manufacturing micro (shaped) electrodes, micro-holes quality, machining time, aperture roundness less than 1um, and high aspect ratio.
  4. The coal oil or EDM dielectric fluid was used in Micro-EDM's machining, which can be added manually or by drop supply.
  5. Tool wear is the difference between the processing before and after, which means the tool wear length.
  6. The feed depth of the spindle includes tool wear and the machining depth of the workpiece, the tool wear caused by a difficult chip removal in deep holes.

For the part of Other comments:

  1. Line 84 on page 2: the sentence has been checked on line 83.
  2. Line 101-102 on page 4: the sentences have been checked from line 102 to 103.
  3. Line 239-244 on page 10: has been rewritten from line 234 to 248。
  4. Last section has been renamed “Conclusion”
  5. Changed most of the desktop micro-EDM machine figures.
  6. Figure 12: has been changed to EDM

Thank you for your precious comments and feedbacks.

I shall be appreciated if you can accept my revision and look forward to hearing from you soon

Sincerely yours,

Yung-yi WU

Taiwan

Reviewer 3 Report

The idea of EDM in particular is to use much softer tool material, however, the authors have used a harder material (WC) to cut another hard material (WC-Co). What are the advantages of this? This seems to be expensive as compared to traditional EDM/micro EDM experiments

Author Response

Dear reviewers,

Thank you for your patience in reviewing and giving comments for corrections of my paper. In respond to your advices, my revision is submitting as follows:

  1. The tungsten cemented carbide (WC-Co) is widely used in the industry and is not easy to cut and process. Tungsten carbide (WC) can provide tool electrodes in the existing market to effectively drill and cut tungsten cemented carbide (WC-Co).
  2. Compared with the traditional EDM/micro EDM experiment, the comparison of cost can be obtained by network information so far. I am waiting for the commercial companies price quotation which will be more comparison and contrast.

Thank you for your precious comments and feedbacks.

I shall be appreciated if you can accept my revision and look forward to hearing from you soon

Sincerely yours,

Yung-yi WU

Taiwan

Round 2

Reviewer 1 Report

The authors have done required revisions. Some minor changes need to be done:

  1. Fig. 2 is very large and out of proportion. Figure size should be shortened.
  2. Fig. 3 needs to be labelled properly.

Author Response

Dear reviewers,

Thank you for your patience in reviewing and giving comments for corrections of my paper. In respond to your advices, my revision is submitting as follows:

  1. Regarding the question of how to measure roundness: It is calculated based on the difference between the horizontal diameter and the vertical diameter. This has been revised and added from line 191.
  2. The Figure 1 has been revised and been changed number 1 into number 2. The sketch also has been added in Figure 1 as your comment.
  3. Scale bars have been added in Figure 4 and Figure 5 ( the original Figures 3 and 4).
  4. Figures 5 and 7 have been labelled and been changed number to 6 and 8, with regard to adding sketches, it shows in Figure 1.
  5. In Fig. 9(10), the diameters are more than 100 um or less than 50um, which is to illustrate the possibility on commercial electrodes and WEDG-grind electrodes being used and completed for basic drilling machining in this micro-ECM system. I am sorry for my negligence for this point where I did not present the single-cut edge electrode at a depth of 1mm with a depth-to-diameter ratio of 70μm for drilling. Thank you very much for reminding of me. Figures 17 and 18 have been complemented.
  6. The Error bars of Figures 11 and 12 have been added.

Thank you for your precious comments and feedbacks.

I shall be appreciated if you can accept my revision and look forward to hearing from you soon

Sincerely yours,

Yung-yi WU

Taiwan
